# Mosquito Vector Competence for Japanese Encephalitis Virus

**DOI:** 10.3390/v13061154

**Published:** 2021-06-16

**Authors:** Heidi Auerswald, Pierre-Olivier Maquart, Véronique Chevalier, Sebastien Boyer

**Affiliations:** 1Virology Unit, Institut Pasteur du Cambodge, Institut Pasteur International Network, Phnom Penh 120210, Cambodia; 2Medical and Veterinary Entomology Unit, Institut Pasteur du Cambodge, Institut Pasteur International Network, Phnom Penh 120210, Cambodia; pomaquart@pasteur-kh.org (P.-O.M.); sboyer@pasteur-kh.org (S.B.); 3Epidemiology and Public Health Unit, Institut Pasteur du Cambodge, Institut Pasteur International Network, Phnom Penh 120210, Cambodia; veronique.chevalier@cirad.fr; 4UMR ASTRE, CIRAD, INRA, Université de Montpellier, 34000 Montpellier, France; 5Institut Pasteur, 75015 Paris, France

**Keywords:** Japanese encephalitis virus, mosquito, vector competence

## Abstract

Japanese encephalitis virus (JEV) is a zoonotic pathogen mainly found in East and Southeast Asia and transmitted by mosquitoes. The objective of this review is to summarize the knowledge on the diversity of JEV mosquito vector species. Therefore, we systematically analyzed reports of JEV found in field-caught mosquitoes as well as experimental vector competence studies. Based on the investigated publications, we classified 14 species as confirmed vectors for JEV due to their documented experimental vector competence and evidence of JEV found in wild mosquitoes. Additionally, we identified 11 mosquito species, belonging to five genera, with an experimentally confirmed vector competence for JEV but lacking evidence on their JEV transmission capacity from field-caught mosquitoes. Our study highlights the diversity of confirmed and potential JEV vector species. We also emphasize the variety in the study design of vector competence investigations. To account for the diversity of the vector species and regional circumstances, JEV vector competence should be studied in the local context, using local mosquitoes with local virus strains under local climate conditions to achieve reliable data. In addition, harmonization of the design of vector competence experiments would lead to better comparable data, informing vector and disease control measures.

## 1. Introduction

Japanese encephalitis (JE) is a vector borne zoonosis and one of the world’s leading encephalitic diseases, particularly in the Asia-Pacific region [1]. The disease is endemic in 24 countries in South and Southeast Asia from Pakistan to Japan, northern Australia and Oceania [2], putting more than three billion people at risk of infection. The annual incidence of JE is estimated to be around 69,000 cases [3] but this is likely to be underestimated due to insufficient surveillance systems and the lack of precise diagnostic tools. Based on an estimated annual loss of 709,000 disability-adjusted life years [4], JE has even a higher disease burden than dengue.

JE is caused by the Japanese encephalitis virus (JEV) belonging to the *Flaviviridae* family [5]. The main epidemiological pattern is an enzootic cycle where the virus is transmitted between birds and/or pigs by mosquitoes [6]. Humans and other mammal species like horses serve as dead-end hosts. Recently, direct pig-to-pig transmission by oronasal infection was demonstrated under laboratory conditions [7]. The importance of this vector-free infection route for the maintenance of the JEV epidemiological cycle is substantiated by mathematical modelling using serological data from field investigations [8]. However, mosquitoes are still considered the key players in terms of virus transmission and therefore investigations of their capacity to efficiently pass on JEV are vital for risk assessments and public health recommendations. In contrast to other known flaviviruses, a broad range of mosquito species can theoretically transmit JEV, including several genera such as *Aedes* (*Ae*.), *Anopheles* (*An*.), *Culex* (*Cx*.) and *Mansonia* (*Ma*.) [9]. The main considered JEV vector species are *Cx*. *tritaeniorhynchus*, *Cx*. *vishnui* and *Cx*. *gelidus* [10,11,12].

The distribution of *Cx*. *tritaeniorhynchus* is widespread across Southeast Asia and adjacent tropical areas. It extends to Australia and from the Middle East to Africa [13,14,15]. It also has recently been reported in Greece. This species is considered the main JEV vector and its distribution area coincides with the epidemiological risk area of JE [16]. *Cx*. *vishnui* is also widely distributed in Southeast Asia, and can be found from India in the west, to Japan in the east, South Korea to the north and Indonesia to the south. This species has long been recognized as an important vector for JEV because the females feed primarily on pigs and birds, but also opportunistically bite humans [17]. The reports on the involvement of *Cx*. *gelidus* in JEV transmission are historic [18,19]. Depending on the country, this species is considered a primary or secondary vector of JEV [12].

The work of Oliveira and colleagues [9,20,21] confirmed that *Cx*. *tritaeniorhynchus* and *Cx*. *gelidus* should be considered primary JEV vectors. Additionally, the authors emphasize the important role of *Cx*. *quinquefasciatus* for JEV transmission. Generally, many species are also considered secondary vectors, particularly *Ae*. *japonicus*, *Cx*. *fuscocephala* and *Cx*. *vishnui* [9].

Vector competence in the context of arbovirus transmission describes the ability of a particular insect to become infected, maintain viral replication and transmit a specific virus. The transmission of the pathogen to the vector occurs during a blood meal on an infectious host, establishing a systemic infection leading to an infection of the salivary glands, enabling the virus to be transmitted by blood feeding to another host. To establish a systemic infection, the virus has to pass several physiological barriers, especially in the midgut and the salivary glands of the mosquito. The period from acquiring the virus to becoming infectious is the so-called extrinsic incubation period (EIP). In contrast to vector competence, which describes only the ability for viral transmission, vector capacity expresses the efficiency of the transmission in a specific vector-host relationship in a given environment. Therefore it includes the vector competence itself but also mosquito parameters (density of vectors in relation to the density of the host, proportion of vectors feeding on the host relative to the gonotrophic cycle, survival rate of the vector and EIP) [22].

Besides the infection of female mosquitoes through the bite of an infected host, JEV can also be transmitted horizontally (veneral) [23] or vertically in mosquitoes (transovarial or during oviposition) [24]. However, we exclude a detailed analysis of these findings, as often the transmission capacity of the offspring was not further investigated.

The objective of this study is to perform a systematic review of JEV competence surveys in mosquitoes and categorize the JEV vector species into confirmed and potential vector species. Hence, confirmed vectors are species with proven vector competence and JEV found in field-caught mosquitoes. Potential vectors are species with demonstrated experimental vector competence but missing evidence of JEV presence in field-caught mosquitoes. Additionally, we describe the diversity of JEV vectors, and compare the methods used to assess vector competence of JEV vectors, and the consequences in terms of interpretation of results.

## 2. Materials and Methods

### 2.1. Literature Search

This study was conducted by using the PubMed database (cutoff date 19 April 2021). We included the search entries “(Japanese encephalitis [Title/Abstract]) AND (virus isolation [Title/Abstract])”, “(Japanese encephalitis [Title/Abstract]) AND (vector competence [Title/Abstract])”, “(Japanese encephalitis [Title/Abstract]) AND transmission [Title/Abstract] AND (mosquito [Title/Abstract])”, “(Japanese encephalitis [Title]) AND (vector [Title])”, “(Japanese encephalitis [Title]) AND (mosquitoes [Title])”, and “(Japanese encephalitis [Title]) AND (mosquito [Title])”. The results were analyzed according to the PRISMA guidelines [25]. We only included studies describing JEV detection and/or isolation from field-caught mosquitoes, and experimental vector competence studies.

### 2.2. Mosquito Classification and Taxonomy

According to the classification provided by the Mosquito Taxonomic Inventory [26] some mosquito names were not valid anymore (i.e., declared as synonyms) and had to be updated. Consequently, when this was the case, the name as it appears in the original article is given in parentheses after the updated name. As a matter of clarity, regarding members of the very large and composite *Aedes* genus we followed the classification of Wilkerson et al. [27].

### 2.3. Classification in Confirmed and Potential Vector Species

A mosquito species was considered a confirmed vector if (i) studies demonstrated the successful isolation of JEV from field-caught mosquitoes, and (ii) if artificial infection experiments showed successful transmission. Whereas potential vectors where mosquito species with proven vector competence but evidence of JEV in field-caught mosquitoes was missing. In this review we only discussed the vector competence as the intrinsic ability of a mosquito species (or population) to transmit JEV.

The herein discussed papers used the following terms: infection rate, defined as proportion of mosquitoes with JEV detected in their bodies among all tested/blood-fed mosquitoes including mosquitoes naturally blood fed, experimentally infected or infected on animals; dissemination rate, related to the proportion of mosquitoes with JEV detected in their legs, wings and/or head among all infected/blood-fed mosquitoes; transmission rate, defined as the proportion of mosquitoes with JEV detected in their saliva or salivary glands among all infected mosquitoes. Additionally, the transmission rate in studies demonstrating the transmission to another animal is defined as proportion of JEV positive animals among those exposed to infected mosquitoes, and does not consider the extent of exposure (i.e., time of exposure and number of infected mosquitoes). The period between infection of the mosquitoes and outcome measures is described in days post infection (dpi).

## 3. Results

To describe the diversity of JEV mosquito vectors, we screened 650 publications and included 158 of them in this review (Figure 1) demonstrating JEV detection and/or isolation from field-caught mosquitoes, and/or investigations of the vector competence for JEV. We split the results accordingly into (i) confirmed vector species (Table 1, Figure 2) when their vector competence was proven by experimental infection and transmission, and JEV was found in field-caught mosquitoes, (ii) potential vector species with proven experimental vector competence but no documented JEV in field mosquitoes.

We identified 14 species as confirmed vectors of JEV (Table 1): *Ae. albopictus*, *Ae. vexans*, *Ae. vigilax*, *Armigeres* (Ar.) *subalbatus*, *Cx. annulirostris*, *Cx. bitaeniorhynchus*, *Cx. fuscocephala*, *Cx. gelidus*, *Cx. pipiens*, *Cx. pseudovishnui*, *Cx. quinquefasciatus*, *Cx. sitiens*, *Cx. tritaeniorhyncus* and *Cx. vishnui*, based on reports stating JEV in field-caught mosquitoes and the confirmation of the vector competence by experimental infection (Appendix A).

We classified 11 mosquito species as potential JEV vector based on experimental vector competence studies: *Ae*. *detritus*, *Ae*. *dorsalis*, *Ae*. *japonicus*, *Ae*. *kochi*, *Ae*. *nigromaculis*, *Ae*. *notosciptus*, *An*. *tessellatus*, *Cx*. *tarsalis*, *Cs*. *annulata*, *Cs*. *inornata* and *Ve*. *funerea*.

Furthermore, we found reports for 26 species from five different mosquito genera where JEV was detected and/or isolated from field-caught mosquitoes. This includes *Aedes* (4 species), *Anopheles* (9 species), *Coquilettidia* (1 species), *Culex* (7 species) and *Mansonia* (5 species) mosquitoes. However, for these species the vector competence was not investigated so far.

### 3.1. Confirmed Vectors

*Aedes* (*Stegomyia*) *albopictus* (Skuse, 1894)

*Ae*. *albopictus* was found JEV positive and the virus was isolated from these mosquitoes in Malaysia and in Taiwan [28,30]. An early investigation of the vector competence of a Taiwanese laboratory colony of *Ae*. *albopictus* was not able to detect infection in mosquitoes fed on viremic pigs [161]. However, later investigations with *Ae*. *albopictus* from Taiwan [32] and France [34] were both able to detect virus in saliva of 47% (14 dpi) and in 20–63% of the mosquitoes at 11–13 dpi, respectively. Both studies infected the mosquitoes by feeding them an artificial, infectious blood meal with a high viral load (Taiwan: 10^7^ pfu/mL; France: 10^6^ ffu/mL). A study on the replication capacity of the JEV chimeric vaccine confirmed replication of JEV (wildtype SA-14) in *Ae*. *albopictus* infected either by intrathoracic injection or by feeding an infectious blood meal [162]. A study with Australian *Ae*. *albopictus* was also successful at demonstrating transmission even by using a lower dose of 10^3.5^ TCID50/mL (approx. 10^3.3^ pfu/mL) for infection by feeding an infectious blood meal [33]. Interestingly, an earlier Taiwanese study [31] using several laboratory colonies of *Ae*. *albopictus* originating from different provinces found only one of the three colonies able to transmit JEV to weanling mice (27–39% transmission rate, depending on mosquito infection via intrathoracic injection or artificial blood meal). Also, a study infecting a Chinese laboratory colony of *Ae*. *albopictus* with JEV strains isolated from different bat species was able to detect JEV in the mosquitoes 4–20 dpi but did not investigate the transmission capacity [163].

*Aedes* (*Ochlerotatus*) *vexans* (Meigen, 1830)

JEV was isolated from field *Ae*. *vexans* mosquitoes in Taiwan [28]. Early experimental infections in 1946 on Guam [35] observed successful transmission to infant mice but neither the mosquito infection procedure nor the transmission rate was described in further detail. Another investigation from the same time performed with US mosquitoes was not able to detect any infection and transmission [36].

*Aedes* (*Ochlerotatus*) *vigilax* (Skuse, 1889)

During an investigation of a JE outbreak in northern Queensland, Australia, JEV was isolated from a pooled sample of *Ae*. *vigilax* mosquitoes from the Torres Strait island region [37]. Also, a laboratory colony and field-caught mosquitoes of *Ae*. *vigilax* (mentioned as *Ochlerotatus vigilax*) from North Queensland, were analyzed for their competence to transmit JEV [38]. This study showed that both mosquito populations could get infected with JEV when fed an artificial, infectious blood meal, but surprisingly transmission to mice was only observed with the field-caught mosquitoes and not with the females from the laboratory colony.

*Armigeres* (*Armigeres*) *subalbatus* (Coquillett, 1898)

JEV was isolated from several pooled samples of *Ar*. *subalbatus* caught in Taiwan in 1997 [28], and several times in Yunnan province in China [40,41]. The virus was found in a pool of mosquitoes collected in India in 2011–2013 [39]. A vector competence study with Taiwanese mosquitoes detected virus in the salivary glands in 40% to 88% of the mosquitoes depending on the JEV strain and the time point after infection [32]. The same group also determined that *Wolbachia* infection has no influence on the JEV transmission capacity of *Ar*. *subalbatus* [42].

*Culex* (*Culex*) *annulirostris* Skuse, 1889

During a JE outbreak in 1995 on Badu Island, located north of the Australia mainland and south of Papua New-Guinea, JEV was isolated from several *Cx*. *annulirostris* [43] and later also on other islands of the Torres Strait [37]. Early experiments with mosquitoes from Guam (then named *Cx*. *jepsoni*) were able to demonstrate transmission to mice [35]. An extensive vector competence study with Australian mosquitoes demonstrated JEV transmission to mice via *Cx*. *annulirostris* from a laboratory colony and via two populations of field collected *Cx*. *annulirostris* from Queensland [38]. Van den Hurk and colleagues later also used a laboratory colony of Australian *Cx*. *annulirostris* to prove JEV transmission to flying fox (*Pteropus alecta* Temminck, 1837) as 60% of the exposed flying foxes seroconverted after exposure to infected mosquitoes [44].

*Culex* (*Culex*) *bitaeniorhynchus* Giles, 1901

JEV positive *Cx*. *bitaeniorhynchus* were found in India [39], South Korea [45,46,47,48] and Malaysia [29], and in the latter several isolates were attained from field-caught mosquitoes. Several studies with mosquitoes from India demonstrated the vector competence of this species [49,50,51,164]. All these studies infected the mosquitoes by feeding on viremic young ducks or chickens, and afterwards demonstrated further transmission from blood-fed mosquitoes to naïve ducklings or chicks.

*Culex* (*Culex*) *fuscocephala* Theobald, 1907

JEV was first isolated from *Cx*. *fuscocephala* in Thailand in 1970 [55], and later also detected and/or isolated in Indonesia [57,165], in Sri Lanka in 1987–1988 [52], in Malaysia in 1993 [59], in Taiwan [54,56] and throughout several surveys in India in mosquitoes collected 1985–1987 [58], 1991–1994 [53] and 2011–2013 [39]. Vector competence studies in the 1970s showed successful transmission to young chickens by mosquitoes from Thailand [60] and Taiwan [61] infected under laboratory conditions by feeding on viremic chicks or pigs, respectively.

*Culex* (*Culex*) *gelidus* Theobald, 1901

JEV was detected in *Cx*. *gelidus* collected between 1987–1988 in Sri Lanka [52], Australia [67] and several times in mosquitoes from India [53,62,63,64,65,66,68,69,70]. The role of this species as JEV vector was further confirmed by several studies describing successful virus isolation from field-caught mosquitoes in India [53,58,81], Indonesia [57,76,165], Vietnam [75], Thailand [71,77], Malaysia [19,59,72,73,74,78], Sri Lanka [52] and Australia [79,80]. Gould and colleagues also performed transmission experiments with a Malaysian *Cx*. *gelidus* laboratory colony and observed a transmission rate to young chickens of up to 85% when mosquitoes were infected by biting viremic chicks [19]. However, when mosquitoes were allowed to bite infected horses no further transmission to chickens could be detected [151], indicating early the role of horses as dead-end host in the JEV transmission cycle producing not a high enough viremia to be infectious for mosquitoes. Later, a small amount of Australian field-caught mosquitoes were infected and a single JEV transmission event to a suckling mouse was observed [38]. A study on a laboratory colony of *Cx*. *gelidus* from India investigated viral growth kinetics and found JEV in the saliva at 10 dpi and 14 dpi [82].

*Culex* (*Culex*) *pipiens* Linnaeus, 1758

JEV was detected in *Cx*. *pipiens* during recent outbreaks in South Korea [46,48,84], and Italy [83], and during the re-instated surveillance activities in Shanghai, China [85]. Reports also exist about early isolations in Japan [86], South Korea [87] and China [88]. Several studies investigated the vector competence of *Cx*. *pipiens,* including the subspecies *Cx*. *pipiens pipiens*, *Cx*. *pipiens molestus* and *Cx*. *pipiens pallens* (Appendix A).

A recent study from New Zealand [166] noted successful infection but no transmission. However, this study was not able to show JEV transmission for any of the investigated mosquito species, even for the well-described JEV vector *Cx*. *quinquefasciatus*, making these results questionable. In contrast, others were able to observe JEV transmission from *Cx*. *pipiens* to mice [90], and determine possible transmission by detecting JEV in the saliva of the mosquitoes as early as 7 dpi for a certain JEV isolate from China [94], and at 21 dpi in *Cx*. *pipiens* from the UK [95]. The virus was also found in the saliva of *Cx*. *pipiens* when infected by intrathoracic injection [97]. In this study, the infected mosquitoes were also able to infect newly hatched ducklings at 10 dpi.

For the subtype *Cx*. *pipiens molestus* efficient transmission to mice was shown with a laboratory colony from the US [36] and one from Taiwan [92]. Transmission to young chickens was also obtained with field-caught *Cx*. *pipiens molestus* from Uzbekistan [93].

In addition to the vector competence for *Cx*. *pipiens molestus*, Reeves & Hammon also observed transmission to mice and a chicken with a US laboratory colony of *Cx*. *pipiens pipiens* [36]. This was confirmed by a recent study using recombinant JEV strains and *Cx*. *pipiens pipiens* from France, measuring JEV in the saliva of up to 41% of the infected mosquitoes [34]. A recent study examined the vector competence of a laboratory colony of *Cx*. *pipiens pipiens* from the UK at different temperatures [98]. At 20 °C, no JEV was detected in saliva, whereas at 25 °C transmission seems possible because 90% of the mosquitoes had JEV in their saliva.

JEV was also found in mosquito pools of *Cx*. *pipiens pallens* collected in 2015 in Shandong province, China [89]. Female *Cx*. *pipiens pallens* were used to investigate the role of mosquito defensins [96,167] and of a C-type lectin protein [168] on the JEV infection documenting substantial amounts of JEV in the salivary glands (10 dpi). Doi and colleagues showed that the subtype *Cx*. *pipiens pallens* could be successfully infected with JEV but only when a sufficiently high virus concentration (at least 10^4^ LD50) was fed via an artificial blood meal [169]. They also demonstrated JEV transmission via *Cx*. *pipiens pallens* to lizards and then further to mice [91]. Another study from Japan demonstrated that this subspecies could become infected when fed on young JEV-infected chicken during the peak of the viremic phase [23]. However, a more recent investigation with field-caught mosquitoes from South Korea was not able to observe transmission to young chickens [156], even though the experimental procedure was proven with *Cx*. *pipiens molestus* [93].

*Culex* (*Culex*) *pseudovishnui* Colless, 1957

JEV was isolated in India from *Cx*. *pseudovishnui* from Karnata [102], from Goa [103], and was also detected in several pooled mosquito samples from other areas in India [99,100,101]. Several vector competence studies were performed with *Cx*. *pseudovishnui* (Appendix A). Mosquitoes from a laboratory colony in Japan were successfully infected when fed on chicken with a viremia of at least 10^3^ LD50, whereas a lower viremia was not sufficient to establish an infection in these mosquitoes 10–14 dpi [169]. Later investigations with mosquitoes from India found JEV in the salivary glands when the mosquitoes were infected via feeding on viremic chicks [104], via intrathoracic injection and feeding of an artificial, infectious blood meal [105].

*Culex* (*Culex*) *quinquefasciatus* Say, 1823

JEV was detected in *Cx*. *quinquefasciatus* mosquitoes from India [39] and Vietnam [106]. Isolations were successful from mosquitoes in India [58], Vietnam [75], Thailand [107] and Taiwan [28]. Early vector competence studies with *Cx*. *quinquefasciatus* (sometimes called *Cx*. (*pipiens*) *fatigans* or *Cx*. *pipiens quinquefasciatus*) laboratory colonies from Japan [169] and India [51,110] were able to observe concentration-dependent infection rates, and confirm transmission to young chickens, respectively. Similar to the experiments with *Cx*. *pipiens pallens*, Doi and colleagues also demonstrated JEV transmission via *Cx*. *pipiens fatigans* to lizards and then further to mice [91]. Several studies in the 1940s on laboratory colonies and mosquitoes from Guam [35,108,109,170] demonstrated transmission of a human JEV isolate from Okinawa to infant mice when the mosquitoes were infected by feeding on infectious blood presented on cotton. Reeves and Hammon also used this technique for successfully infecting *Cx*. *quinquefasciatus* and demonstrating further transmission to mice, whereas they were not able to infect mosquitoes successfully by feeding them on viremic chicken [36]. More recent vector competence studies detected JEV in salivary glands two weeks after infection of *Cx*. *quinquefasciatus* from Taiwan [32], and in the saliva of mosquitoes from USA [112,113]. Laboratory colony mosquitoes from China were also successfully infected with JEV isolates from bats [163]. In experiments with Indian *Cx*. *quinquefasciatus* it was also documented that colonization with certain bacteria (*Pseudomonas* sp. and *Acinetobacter junii*) slightly increased their susceptibly for infection with JEV [171], and that simultaneous or sequential infection with Bagaza virus (*Flaviviridae*, Ntaya Flavivirus serocomplex) reduced replication of JEV [172]. An investigation of van den Hurk and colleagues revealed higher infection rates and JEV transmission to mice via laboratory colony mosquitoes, whereas the field-caught mosquitoes were not able to transmit JEV [38]. Not only do the microbiota and origin of the mosquitoes influence their transmission capacity, but also the ambient temperature, as shown for *Cx*. *quinquefasciatus* held either at 23 °C or 28 °C with mosquitoes from Brazil [111]. Infection and dissemination rate were similar but the transmission rate 14 and 21 dpi was elevated at the higher temperature. An extensive study using several JEV strains revealed that viruses belonging to genotype I had higher infection, dissemination and transmission rates in *Cx*. *quinquefasciatus* than viruses from genotype III [114]. Additionally, the EIP was shorter for infections with genotype I strains. However, one study performed in New Zealand with endemic *Cx*. *quinquefasciatus,* as well as mosquitoes from a US laboratory colony, was not able to detect transmission [166].

*Culex* (*Culex*) *sitiens* Wiedemann, 1828

JEV was found in *Cx*. *sitiens* in East and South Asia, as well as in northern Oceania. The virus was found and/or isolated from mosquitoes in Malaysia [29,115], Taiwan [28,116], Papua New Guinea [118] and Australia [67,117,119,120]. In Australia, *Cx*. *sitiens* was identified as a competent JEV vector using a virus strain isolated from Australian *Och*. *vigilax* and mosquitoes from a laboratory colony from Queensland [38].

*Culex* (*Culex*) *tritaeniorhynchus* Giles, 1901

*Cx*. *tritaeniorhynchus* is considered the primary JEV vector in a lot of countries. It was first isolated from this mosquito species in the 1930s in Japan [173] as well as later through several decades of Japanese surveillance [130,136,145]. Virus isolation was also successful with mosquitoes from Indonesia [57,132,133,165], India [53,58,70,102,129,137,143], Malaysia [59,72,74], Thailand [71,135], Taiwan [28,56,131,134], South Korea [139], Vietnam [75,141], China [40,41,89,138,140,142,147], Cambodia [146] and Singapore [144]. In addition, JEV was detected over the last few decades in *Cx*. *tritaeniorhynchus* in Sri Lanka [52], India [39,53,62,63,64,65,66,68,69,99,100,101,121,122,126], Malaysia [29], Taiwan [116], Vietnam [106], South Korea [45,46,47,48,123], Taiwan [54,124], Japan [125] and China [85,127,128]. Simultaneously to the JEV isolation, Mitamura and colleagues also used *Cx*. *tritaeniorhynchus* mosquitoes naturally infected with JEV to demonstrate transmission of JEV to mice [148]. A study on the replication capacity of the JEV chimeric vaccine confirmed replication of JEV (wildtype SA-14) in *Cx*. *tritaeniorhynchus* infected either by intrathoracic injection or by feeding an infectious blood meal [162]. Others were also able to successfully infect mosquitoes from Japan by feeding them on viremic pigs [161] or infecting them via an artificial, infectious blood meal [174], and infect mosquitoes from a laboratory colony in Taiwan [31]. Vector competence studies showed successful JEV transmission to mice [23,90,92,148,152], horses [151], pigs [150,152], young chickens [51,60,61,110,149,151,154,156], young ducks [49] as well as several ardeid birds like Black-crowned Night Herons, Plumed Egrets, Great Egrets [150] and Indian pond herons [154]. Most of the early studies infected the respective mosquitoes by feeding them on viremic pigs or chickens [49,51,60,61,110,150,151,152,154,156,161,169]. Nowadays vector competence studies use mostly artificial blood meals to infect mosquitoes because the virus titer can be easier modified then in viremic animals used for feeding. This technique was used to demonstrate the vector competence of *Cx*. *tritaeniorhynchus* from Japan [23,153,157], Singapore [149] and Taiwan [92,155]. Another route for infection is the direct injection of JEV into the thorax of a mosquito. This is rarely used for vector competence, as results from this kind of studies cannot elusively document vector competence as the virus is not ingested and therefore has not crossed the mosquito midgut barrier. Several studies using *Cx*. *tritaeniorhynchus* showed that with intrathoracic injection high infection rates can be reached, also leading to high transmission rates [105,155]. The investigation of Chen and colleagues also showed that the JEV vaccine strain J 2-8 is not able to replicate in mosquitoes whereas its parent viral strain SA-14 establishes a disseminated infection resulting in successful JEV transmission to mice [155]. *Cx*. *tritaeniorhynchus* mosquitoes were also used to investigate the influence of Bagaza virus on the replication of JEV [172]. The replication was impaired but the study did not look into the effect on transmission. A recent investigation of a *Cx*. *tritaeniorhynchus* laboratory colony in Japan showed transmission of three different JEV genotypes. Similar to the results obtained with *Cx*. *quinquefasciatus* [114], JEV genotype I had a shorter EIP in *Cx*. *tritaeniorhynchus* compared to genotypes III and V, whereas the transmission rates where similar for all three tested genotypes [157]. 

*Culex* (*Culex*) *vishnui* Theobald, 1901

JEV was detected in *Cx*. *vishnui* in Malaysia [29], Vietnam [141], and several times in India [53,62,63,100,101,143,158,159]. Additionally, the virus was isolated from mosquitoes in India [53,160], Thailand [135] and Indonesia [57]. A laboratory colony of Indian *Cx*. *vishnui* was also used to prove the vector competence of this species with a JEV strain isolated in India [105].

### 3.2. Potential Vectors

Eleven mosquito species, belonging to five genera, were successfully tested under laboratory conditions to transmit JEV, but the virus was neither detected nor isolated from field-caught mosquitoes. The species fulfilling these criteria are *Ae*. *detritus*, *Ae*. *dorsalis*, *Ae*. *japonicus*, *Ae*. *kochi*, *Ae*. *nigromaculis*, *Ae*. *notosciptus*, *An*. *tessellatus*, *Cx*. *tarsalis*, *Cs*. *annulata*, *Cs*. *inornata* and *Ve*. *funerea*.

*Aedes* (*Ochlerotatus*) *detritus* (Haliday, 1833)

*Ae*. *detritus* (*Ochlerotatus detritus*) mosquitoes from England, were investigated for their potential to transmit JEV at 23 °C and 28 °C [111]. Infection and dissemination rates were similar, but the transmission was more efficient at the lower temperature (67% for 23 °C vs. 33% for 28 °C at 21 dpi). However, the number of investigated mosquitoes was also very low. This finding is lower than *Cx*. *quinquefasciatus* transmission rates observed in the same study at elevated temperatures (70% for 28 °C vs. 50% for 23 °C at 21 dpi).

*Aedes* (*Ochlerotatus*) *dorsalis* (Meigen, 1830)

In 1946, Reeves and Hammon published their findings on the JEV transmission by North American *Ae*. *dorsalis* [36]. Mosquitoes were infected by feeding them with an artificial, infectious blood meal. Sixteen days later, they were allowed to bite young mice. One of the six mice used in this experiment was successfully infected with JEV from the bite of *Ae*. *dorsalis*. 

*Aedes* (*Hulecoeteomyia*) *japonicus* (Theobald, 1901)

*Ae*. *japonicus* was tested several times for its capacity to transmit JEV. Experiments with *Ae. japonicus* from Japan [23] were able to document transmission to mice when they infected the mosquitoes by feeding on viremic chicken displaying a low viral titer (10^3.7^ pfu/mL) as well as by feeding them an artificial blood meal with a high viral dose (10^6.2^ pfu/mL). Huber and colleagues showed that *Ae*. *japonicus* collected in Germany can be successfully infected with JEV [175]. A recent study with a laboratory colony of *Ae*. *japonicus* from Japan demonstrated the successful infection, dissemination and transmission of three different JEV genotypes [157].

*Aedes* (*Ochlerotatus*) *kochi* (Dönitz, 1901)

A large study investigating the JEV vector capacity of 16 mosquito species from Australia included *Ae*. *kochi* and showed that one of the two field-collected mosquito populations was able to transmit JEV [38]. Although the number of investigated mosquitoes was very low, the transmission to mice was observed only with a single mosquito.

*Aedes* (*Ochlerotatus*) *nigromaculis* (Ludlow, 1906)

In the early 1940s, *Ae*. *nigromaculis* mosquitoes from the US were orally infected via an artificial blood meal [36]. Successful infection was proven by recovering the virus from blood-fed mosquitoes 16 days after infection. Additionally, this study demonstrated JEV transmission to mice as several mice developed encephalitis and the virus was cultivated from brain samples of these mice.

*Aedes* (*Rampamyia*) *notoscriptus* (Skuse, 1889)

Van den Hurk and colleagues also investigated an *Ae*. *notoscriptus* (mentioned as *Ochlerotatus notoscriptus*) laboratory colony, and field-caught mosquitoes from Queensland, Australiafor their JEV vector competence [38]. From laboratory colony mosquitoes, they detected transmission to three of the eleven mice in their experiment. As seen with other field-caught populations in their study, the initial number of infected mosquitoes and their survival rate was very low, and the field mosquitoes were not probing on the infant mice therefore the transmission could not be tested but only infection and dissemination of JEV. Similar problems were seen with field-caught *Ae*. *notoscriptus* from New Zealand [166]. This study was not even able to detect infection in their mosquitoes two weeks after the infectious blood meal. Therefore, the vector competence status of *Ae. notoscriptus* remains questionable.

*Anopheles* (*Anopheles*) *tessellatus* Theobald, 1901

A laboratory strain of *An*. *tessellatus* was shown to be able to transmit JEV to young chickens, the mosquitoes being infected also by feeding on viremic chicks [110]. This is the only study proving JEV vector competence for an *Anopheles* species. Other studies on *An*. *hyrcanus* [176] and *An*. *freeborni* [36] were not able to demonstrate transmission to chicks or mice.

*Culex* (*Culex*) *tarsalis* Coquillett, 1896

A study using North American *Cx*. *tarsalis* infected with JEV via an artificial blood meal was performed [36]. The infection of the mosquitoes was proven by detecting the virus from blood-fed mosquitoes more than two weeks after the infectious blood meal. Furthermore, transmission to young mice was also demonstrated.

*Culiseta* (*Culiseta*) *annulata* Schrank, 1776

In a lab survey performed in the UK [95], *Cs*. *annulata* was shown to be able to transmit JEV, but only when the mosquitoes were kept at 21 °C, whereas when held at 24 °C no virus was detectable in their saliva. 

*Culiseta* (*Culiseta*) *inornata* Williston, 1893

As with other US mosquitoes, Reeves and Hammon also tested *Cs*. *inornata* for its JEV vector competence [36]. Infection was demonstrated by recovering the virus from blood-fed mosquitoes after artificial blood meal, and the transmission to young mice was shown too. In contrast, *Cs*. *incidens* could also get infected with JEV but was not able to transmit the virus in return [36].

*Verrallina* (*Verrallina*) *funerea* Theobald, 1903

Besides other Australian mosquito species, field-caught *Ve*. *funereal* mosquitoes from North Queensland were shown to be able to transmit JEV to young mice when infected via an artificial blood meal [38].

### 3.3. Mosquito Species with JEV Isolation in the Field

Additionally, to confirmed and potential vectors there are mosquito species with documented JEV detection and/or isolation from field mosquitoes (Table 2).

Seven other mosquito species were demonstrated either to be unable to transmit the virus or their JEV vector competence was not tested so far. However, reports on JEV isolations from field-caught mosquitoes of these species imply their possible role as JEV vectors (Table 2). JEV strains were isolated from *Ae*. *butleri* and *Ae*. *lineatopennis* in Malaysia [29,59]. Also the isolation of JEV from various *Anopheles* species was several times successful: from *An*. *annularis* and *An*. *vagus* in Indonesia [57,133], from *An*. *sinensis* in Yunnan province in China [40,41,142], from *An*. *subpictus* [102], and from *An*. *peditaeniatus* [58] in India, and from diverse *Anopheles* as well as *Mansonia* species in Malaysia [74]. Finally, the virus was also isolated from *Cx*. *annulus* and *Cx*. *fuscanus* in Taiwan [28,56,131,177,178], and from *Coquillettiddia ochracea* from Shandong province, China [89].

**Table 2 viruses-13-01154-t002:** Mosquito species with evidence for JEV from field-caught mosquitoes.

Mosquito Species	Virus Detection in Field-Caught Mosquitoes	Virus Isolation from Field-Caught Mosquitoes
***Aedes butleri***		1992 in Malaysia [29]; 1992–1993 in Malaysia [59]
***Aedes curtipes***		1968 in Malaysia [74]
***Aedes lineatopennis***		1992–1993 in Malaysia [59]
***Aedes vexans nocturnus***	2000–2004 in Taiwan [116]	
***Anopheles* ssp**		1969 in Malaysia [74]
***Anopheles annularis***		1978–1980 in Indonesia [57]; 1979 in Indonesia [133]
***Anopheles barbirostris***	1973 in India [158]; 2011–2013 in India [39]	
***Anopheles hyrcanus***	1973 in India [158]; 1974–1975 in India [179]	
***Anopheles pallidus***	2011–2013 in India [39]	
***Anopheles peditaeniatus***		1985–1987 in India [58]
***Anopheles subpictus***	1996 in India [121]; 1997–1999 in India [180]; 2011–2013 in India [39]	1977–1979 in India [102]
***Anopheles sinensis***		2007 in China [40]; 2007–2009 in China [142]; 2009–2010 in China [41]
***Anopheles vagus***		1978–1980 in Indonesia [57]; 1979 in Indonesia [133]
***Coquillettidia ochracea***		2015 in China [89]
***Culex annulus***		1967 in Taiwan [177]; 1969 in Taiwan [131,178]; 1974–1976 in Taiwan [56]; 1995–1996 in Taiwan [28]
***Culex epidesmus***	1974–1975 in India [179]	
***Culex fuscanus***		1995–1996 in Taiwan [28]
***Culex infula***	2011–2013 in India [39]	
***Culex orientalis***	2012 in South Korea [84]	
***Culex rubithoracis***	2002–2004 in Taiwan [116]	
***Culex whitmorei***	1962–1966 in India [129]; 1987–1988 in Sri Lanka [52]	
***Mansonia* ssp**		1969 in Malaysia
***Mansonia bonneae/dives***		1969 in Malaysia [74]
***Mansonia annulifera***	1999–2000 in India [65]; 2011–2013 in India [39]	
***Mansonia indiana***	1996 in India [121]; 1999–2000 in India [65]	
***Mansonia uniformis***	1987–1988 in Sri Lanka [52]; 1996 in India [121]; 1999–2000 in India [65]; 2011–2013 in India [39]	1969 in Malaysia [74]

## 4. Discussion

This article mainly highlights the diversity of mosquito species able to transmit JEV and the diversity of methodology used for vector competence experiments.

### 4.1. Diversity of Mosquito Vector Species and Consequence in Terms of Public Health

Based on our literature search we found 14 mosquito species as confirmed vectors and eleven species we considered as potential vectors. An earlier meta-analysis on JEV infection on vectors and hosts [20] highlighted the importance of the *Culex* genus as important JEV vectors. However, the highest susceptibility (measured as minimum infection rate) was found for *An*. *subpictus*. Overall, the highest JEV infection rates were reported in *Cx*. *annulirostris*, *Cx*. *sitiens*, *Cx*. *fuscocephala* and *Ae*. *japonicus* [21]. A recent study highlighted the importance of *Cx*. *tritaeniorhynchus*, *Cx*. *gelidus*, *Cx*. *sitiens* and *Cx*. *fuscocephala* as JEV vector species [9,21], which is in accordance with these species categorized as confirmed vector species. Generally, *Cx*. *tritaeniorhynchus* is acknowledged as an important vector species, and is certainly studied the most (Appendix A) [11,38,181]. In contrast, *Cx*. *vishnui* was found positive in the field several times in different countries (Table 1), but only one study confirmed its capability to transmit the virus [105]. This could be partially explained by the difficulty to rear this species and subsequently the issues to perform vector competence studies.

All 14 mosquito species described here as competent vectors are also known to bite pigs and humans, and are generally well adapted to live in close proximity to humans and human settlements. The biting behavior is an important component of the vectorial capacity of a vector species. For *Ae*. *albopictus*, *Ae*. *vexans* and *Ar*. *subalbatus* [182,183,184] it is well documented that they bite birds, mammals (including pigs) and humans. *Culex* mosquitoes are generally described as ornithophilic species meaning that they prefer to bite birds. The confirmed JEV vectors, *Cx*. *annulirostris*, *Cx*. *bitaeniorhynchus*, *Cx*. *gelidus*, *Cx*. *fuscocephala*, *Cx*. *pipiens*, *Cx*. *quinquefasciatus*, *Cx*. *sitiens*, *Cx*. *tritaeniorhyncus* and *Cx*. *vishnui* are mainly opportunistic feeders and were reported to bite both birds and mammals [12,185,186,187,188,189,190]. Moreover, *Cx. pipiens* and *Cx*. *quinquefasciatus* seem to prefer feeding on humans rather than birds [189,191,192,193]. Also, *Cx*. *gelidus*, *Cx*. *tritaeniorhynchus* and *Cx*. *vishnui* prefer large mammals (pigs and cows) over birds as shown in a large field study investigating the trophic behavior of Cambodian mosquitoes under natural conditions [191]. The host feeding behavior combined with the competence for JEV transmission of these vectors should be issued in risk assessment studies.

One of the first public health issues is how well the combination virus-vector can adapt to weather conditions different from its current geographic range. Indeed, the increasingly frequent and rapid population movements often participate in emergence or re-emergence of viruses. In particular, with species such as *Ae*. *vexans* or *Cx*. *pipiens,* well adapted to temperate climates, there is a real risk of seeing JE emerging in temperate countries. In the Southern hemisphere, *Cx*. *annulirostris* is now spreading further and further South [194] or *Ae*. *albopictus* invades habitats in Southern and Central Europe, as well as temperate regions in the United States and recently in Canada [195,196,197]. In addition, the main JEV vector *Cx*. *tritaeniorhynchus* is present across South, East and Southeast Asia, demonstrating its high adaptability among different habitats and climate zones [198,199,200].

This review also highlights one of the recurring issues related to the discipline of medical entomology, and to taxonomy in general. In fact, on several occasions, we have come up against the difficulty of finding valid species names (*Cx*. *molestus* and *Och*. *vivax* for *Cx*. *pipiens* and *Ae*. *vexans*, respectively). Even in a recent meta-analysis [20], discrepancies can be found. Mosquito taxonomy is always changing and keeping up with actual and valid names can be a daunting task. As an example, the name of *Aedes albopictus* is widely known by a large part of the scientific and public communities and stakeholders, while its recent change to *Stegomyia albopicta* creates communication problems since people might think that it is another species. While adapting and updating taxonomical names is important to try fitting a phylogenetic reality, taxonomists also have to bear in mind that changing names can potentially alter public health communication.

### 4.2. Diversity of Vector Competence Experiments: Problems and Solutions

For some vector species, it is clearly established that they are competent vectors for JEV as the virus was detected and/or isolated several times form this species and transmission was observed in various experimental infection experiments. In this review, we identified *Ae*. *albopictus*, *Ae*. *vexans*, *Ae*. *vigilax*, *Ar*. *subalbatus*, *Cx*. *annulirostris*, *Cx*. *bitaeniorhynchus*, *Cx*. *fuscophala*, *Cx*. *gelidus*, *Cx*. *pipiens*, *Cx*. *pseudovishnui*, *Cx*. *quinquefasciatus*, *Cx*. *sitiens*, *Cx*. *tritaeniorhynchus* and *Cx*. *vishnui* as confirmed vectors. However, for some species there are contradictory results regarding their potential of JEV transmission. Discrepancies in the outcome of vector competence studies might be caused by differences in the infection method, the mosquito population or the virus used for infection. Vector competence studies have several practical restrictions, leading to a broad variety of experimental designs that were used over the past decades. Therefore, the experimental determination of vector competence poses several challenges, and a broad range of mosquitoes, JEV strains and infection methods can be used in the different studies investigating the vector competence.

#### 4.2.1. Influence of Mosquito Origin and Rearing on Vector Competence

The mosquitoes that can be used for the studies range from established colonies, long adapted to laboratory conditions, to early generations (F1–F5) of field-caught mosquitoes. The origin of the mosquitoes itself can influence the outcome of the vector competence studies as was nicely demonstrated with a laboratory colony of *Cx*. *annulirostris* and *Cx*. *quinquefasciatus* able to transmit JEV, whereas field-caught mosquitoes of the same species showed limited or no transmission [38]. However, mosquitoes always have to be reared at least for one generation in the laboratory, as wild adult mosquitoes can be collected to get eggs from females or by directly collecting eggs and larvae in breeding sites. Therefore, even if F1 mosquitoes are often used for vector competence studies, the rearing under laboratory condition might influence their potential for virus transmission. Several studies have shown that the environmental conditions [153,201], and therefore also the rearing in the laboratory, can influence the transmission of JEV in certain mosquito species. The temperature is an especially important parameter as seen in comparative vector competence studies [95,98,111].

#### 4.2.2. Influence of Virus Strain on Vector Competence

The virus used for the vector competence study can dramatically influence their outcome. In the early days of JEV research, virus strains were isolated by inoculating suckling mice intracranial. This method favors the isolation of neurotropic virus strains. The isolates were then passaged several times in mice, which might lead to further adaptation of the virus strains. Nowadays, arboviruses are mostly isolated in cell culture using mosquito cell lines. However, since decades, the most commonly used cell line for arbovirus isolation is C6/36 [202], which is a cell line adapted from larvae of *Ae*. *albopictus*. This could also introduce a bias, especially for viruses where *Aedes* species are not the primary vectors as it is the case for JEV. It was shown that JEV attenuation through several cell culture passages can lead to loss of infectivity in mosquitoes [203]. Additionally, the viral titer used for infecting mosquitoes is a crucial bottleneck for proving vector competence. Many studies use a rather high viral load (above 10^5^ infectious units/mL; see Appendix A). It is questionable if high infection titers represent the biological situation properly as peak viremia titers are reported for pigs with 10^3^ to 10^5^ TCID50/mL [150,204,205] and for young poultry with 10^4^ to 10^6^ pfu/mL [49,206] in experimental studies. This issue is also documented by the work of Gould and colleagues on *Cx*. *gelidus* showing that mosquitoes fed on viremic young chicken could successfully transmit JEV but not if the mosquitoes fed on viremic horses [151]: horses are dead-end-hosts and therefore do not develop sufficiently high viremia to infect mosquitoes. The influence of the virus strain used for vector competence studies was also observed, showing higher infection, dissemination and transmission rates, as well as a shorter EIP for JEV genotype I strains than for genotype III isolates [114]. A shorter EIP for JEV genotype I was also observed with *Cx*. *tritaeniorhynchus* [157].

#### 4.2.3. Influence of Applied Techniques on Vector Competence

The infection methodology and the outcome measurement are of importance. The first experimental infection and transmission study was published in 1936 in Japan with naturally JEV-infected *Cx*. *tritaeniorhynchus* [148]. Besides this initial study, later investigations infected mosquitoes under laboratory conditions. Nowadays, infection by feeding mosquitoes with artificial, infectious blood meals is the most common method, whereas early studies often let the mosquitoes feed on viremic animals like pigs or young chickens or ducks. The latter mimics the natural infection process; however, it creates the need to handle both viremic animal(s) and mosquitoes under special containment conditions (often biosafety level 3, BSL3). These restricted conditions drastically limit the number of institutions able to perform these experiments. Mosquitoes might show preferences for feeding on the blood of certain species (e.g., rodent blood often used due to availability and feeding preference of mosquitoes). All these factors can influence the outcome of the vector competence study. A standardization of the infection method appears a very challenging task, especially due to the varying feeding and host preferences of different mosquito species. Besides differences in the infection method, the outcome of infection is determined in various ways. Three different parameters can be measured by classical vector competence studies: infection, dissemination and transmission. For novel or unknown vector species, the EIP is an important parameter that should be determined by sampling the infected mosquitoes over a broad range of time points (from 7 dpi up to the death of the mosquitoes). However, most studies used 14 and/or 21 dpi as preferred time point(s).

#### 4.2.4. Other Factors Influencing Vector Competence

Additionally, the vector competence might be influenced by local adaption mechanisms between vector and virus [205]. Despite the challenges of these experimental studies, the investigation of the vector competence of local mosquitoes to local arbovirus strains taking local conditions into account, as shown for other arboviruses with e.g., temperature fluctuations [207,208,209], is important, as it provides valuable data for risk assessments concerning the spread and (re)emergence of JEV [210,211,212]. The influence of the microbiome [213] of the local mosquito populations on the vector competence should be investigated. Increased focus on the influence of insect-specific viruses revealed their often impairing effect on virus transmission [214]. For instance, it was shown that co-infection with the Banna virus M14 (*Reoviridae*) decreased the infectivity of JEV in *Cx*. *tritaeniorhynchus* dramatically [215]. In addition, the effect of infection with *Wolbachia* endobacteria should be considered, as they are widespread in several JEV vector species like *Ae*. *albopictus*, *Cx*. *quinquefasciatus* and *Ar*. *subalbatus* [42,216,217]. This is of particular interest as the impact of trans-infection of JEV vector species with non-naïve *Wolbachia* strains is currently under investigation as a measure for vector control [218].

The broad diversity in the experimental design, as well as used mosquitoes and JEV strains, can be easily seen for mosquito species with many studies such as *Cx*. *quinquefasciatus* and *Cx*. *tritaeniorhynchus* (Appendix A). Whereas the studies nearly always come to the conclusion that these species are competent JEV vectors, the level of transmission varies greatly between the studies. As the studies used very different methods to investigate the vector competence, the detailed outcome of these experiments are hardly comparable.

In addition, the transmission capacity on specific ecological niches should not be neglected. Some studies investigated JEV transmission, including bats in its epidemiological cycle. It was shown that *Ae*. *albopictus* and *Cx*. *quinquefasciatus* can be infected with JEV strains found in bats [163], and that *Cx*. *annulirostris* can transmit the virus to flying foxes (*Pteropus alecto*) [44]. Lizards were also demonstrated to be competent hosts for JEV and supporting JEV transmission via *Cx*. *pipiens pallens* and *Cx*. *quinquefasciatus* [91].

Many studies were performed in countries that have the necessary resources (laboratory capacity, financial support, trained personnel) like Japan, Australia, USA, Taiwan, India or South Korea (Appendix A). This is dangerous as there is a lack of data for many of the developing countries where JEV is endemic leading to neglected JEV awareness, preparedness and control.

Finally, survival rate upon JEV infection should also be carefully monitored, as a certain amount of mosquitoes dies from JEV infection [157]. This effect of JEV on the lifespan of the mosquitoes is important to consider when studying the vector competence and estimating the risk of transmission.

## 5. Conclusions

We described the variety of species able to transmit JEV successfully and discussed the broad variety of vector competence studies and the challenges that come with them. Overall, the risk assessment on potential vectors should always include information on the abundance, spatial and temporal distribution of the mosquito species, as well as surveillance of wild mosquitoes for the presence of JEV, and not only be based on vector competence experiments performed under controlled conditions in the laboratory. The JEV vector competence should preferably be studied in the local context, infecting local mosquitoes with local viral strains under local climate conditions to achieve reliable data. In addition, harmonization of the design of vector competence investigations would lead to better comparable data, informing vector and disease control measures.

## Figures and Tables

**Figure 1 viruses-13-01154-f001:**
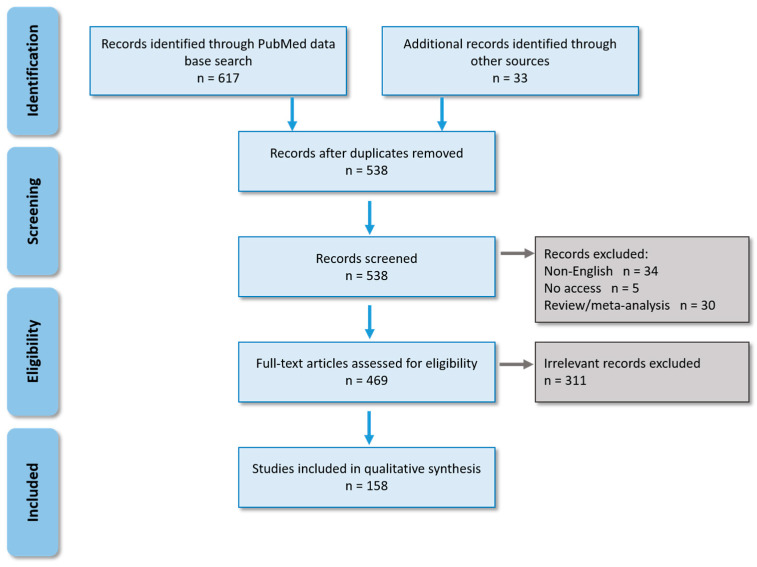
PRISMA flow chart [25] for systematic search of relevant publications.

**Figure 2 viruses-13-01154-f002:**
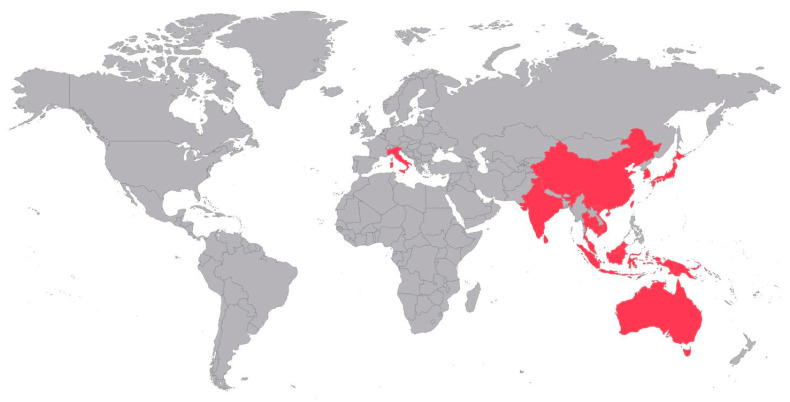
Distribution of reports on JEV detection and isolations from field-caught mosquitoes of the confirmed vector species *Ae*. *albopictus*, *Ae*. *vexans*, *Ae*. *vigilax*, *Ar*. *subalbatus*, *Cx*. *annulirostris*, *Cx*. *bitaeniorhynchus*, *Cx*. *fuscocephala*, *Cx*. *gelidus*, *Cx*. *pipiens*, *Cx*. *pseudovishnui*, *Cx*. *quinquefasciatus*, *Cx. sitiens*, *Cx*. *tritaeniorhyncus* and/or *Cx*. *vishnui*.

**Table 1 viruses-13-01154-t001:** Confirmed JEV mosquito vector species.

Mosquito Species	Virus Detection in Field-Caught Mosquitoes	Virus Isolation from Field-Caught Mosquitoes	Proven Vector Competence *
***Aedes albopictus***		1995–1996 in Taiwan [28]; 1992 in Malaysia [29]; 2005–2012 in Taiwan [30]	[31,32]; [33,34]
***Aedes vexans***		1995–1996 in Taiwan [28]	[35,36]
***Aedes vigilax***		1997–1998 in Australia [37]	[38]
***Armigeres subalbatus***	2011–2013 in India [39]	1995–1996 in Taiwan [28]; 2007 in China [40]; 2009–2010 in China [41]	[32,42]
***Culex annulirostris***		1995 in Australia [43]; 1997–1998 in Australia [37]	[35]; [38,44]
***Culex*** ***bitaeniorhynchus***	1992 in Malaysia [29]; 2008–2010 in South Korea [45]; 2008–2010 in South Korea [46]; 2010 in South Korea [47,48]; 2011–2013 in India [39]	1992 in Malaysia [29]	[49,50,51]
***Culex*** ***fuscocephala***	1987–1988 in Sri Lanka [52]; 1991–1994 in India [53]; 2006–2008 in Taiwan [54]; 2011–2013 in India [39]	1970 in Thailand [55]; 1974–1976 in Taiwan [56]; 1978–1980 in Indonesia [57]; 1985–1987 in India [58]; 1991–1994 in India [53]; 1992 in Malaysia [59]	[60,61]
***Culex*** ***gelidus***	1987–1988 in Sri Lanka [52]; 1991–1994 in India [53]; 1995–1997 in India [62]; 1996–2004 in India [63]; 1998–2000 in India [64]; 1999–2000 in India [65]; 2002–2005 in India [66]; 2003–2004 in Australia [67]; 2010–2013 in India [68]; 2011 in India [69]; 2016 in India [70]	1954–1956 in Malaysia [19]; 1962–1968 in Thailand [71]; 1968–1969 in Malaysia [72]; 1969 in Malaysia [73,74]; 1972–1973 in Vietnam [75]; 1972–1974 in Indonesia [76]; 1978–1980 in Indonesia [57]; 1985–1987 in India [58]; 1986–1987 in Thailand [77]; 1987–1988 in Sri Lanka [52]; 1991–1994 in India [53]; 1992 in Malaysia [29]; 1992–1994 in Malaysia [78]; 2000 in Australia [79,80]; 2002–2006 in India [81]	[19,38,82]
***Culex*** ***pipiens*** **(including subspecies** ***molestus*, *pallens*, *pipiens*)**	2008–2010 in South Korea [46]; 2010 in South Korea [48]; 2010–2011 in Italy [83]; 2012 in South Korea [84]; 2016 in China [85]	1952–1957 in Japan [86]; 1960s in South Korea [87]; 1970s in China [88]; 2015 in China [89]	[34,36,90,91,92,93,94,95,96,97,98]
***Culex*** ***pseudovishnui***	1985 + 1987 in India [99]; 2009–2010 in India [100]; 2010–2013 in India [101];	1977–1979 in India [102]; 1988 in India [103]	[104,105]
***Culex*** ***quinquefasciatus***	2009 in Vietnam [106]; 2011–2013 in India [39]	1972–1973 in Vietnam [75]; 1985–1987 in India [58]; 1995–1996 in Taiwan [28]; 2003 in Thailand [107]	[32,35,36,38,51,91,108,109,110,111,112,113,114]
***Culex*** ***sitiens***	1992 in Malaysia [29,115]; 2002–2004 in Taiwan [116]; 2003–2005 in Australia [67]; 2004 in Australia [117]	1992 in Malaysia [29]; 1995–1996 in Taiwan [28]; 1997–1998 in Papua New Guinea [118]; 1998 in Australia [119]; 2000 in Australia [120]	[38]
***Culex*** ***tritaeniorhynchus***	1985–1987 in India [99]; 1987–1988 in Sri Lanka [52]; 1991-1994 in India [53]; 1992 in Malaysia [29]; 1995–1997 in India [62]; 1996 in India [121]; 1996–2004 in India [63]; 1998–2000 in India [64]; 1999–2000 in India [65]; 2001–2003 in India [122]; 2002–2004 in Taiwan [116]; 2002–2005 in India [66]; 2006–2008 in Taiwan [54]; 2008–2009 in South Korea [123]; 2008–2010 in South Korea [46]; 2008–2010 in South Korea [45]; 2009 in Vietnam [106]; 2009–2010 in India [100]; 2009–2010 in Taiwan [124]; 2010 in South Korea [47,48]; 2010–2013 in India [68,101]; 2010–2014 in Japan [125]; 2011 in India [69]; 2011–2013 in India [39]; 2011–2014 in India [126]; 2013 in China [127]; 2016 in China [85]; 2016 in India [70] 2018 in China [128]	1962–1966 in India [129]; 1962–1968 in Thailand [71]; 1964–1968 in Japan [130]; 1968–1969 in Malaysia; 1968–1970 in Malaysia [74] 1969 in Taiwan [131] 1972 in Indonesia [132]; 1972–1973 in Vietnam [75]; 1974–1976 in Taiwan [56]; 1977–1979 in India [102]; 1978–1980 in Indonesia [57]; 1979 in Indonesia [133]; 1980s in Taiwan [134]; 1982 in Thailand [135]; 1985–1987 in India [58]; 1986–1991 in Japan [136]; 1990s in India [137]; 1991–1994 in India [53]; 1992–1993 in Malaysia [59]; 1995–1996 in Taiwan [28]; 2000s in China [138]; 2000 in South Korea [139]; 2005–2006 in China [140]; 2006–2008 in Vietnam [141]; 2007 in China [40]; 2007–2009 in China [142]; 2009–2010 in China [41]; 2011 in India [143]; 2011–2013 in Singapore [144]; 2013 in Japan [145]; 2014 in Cambodia [146]; 2015 in China [89]; 2015 in China [147]	[23,49,51,60,61,90,92,105,110,148,149,150,151,152,153,154,155,156,157]
***Culex*** ***vishnui***	1970s in India [158]; 1991–1994 in India [53]; 1992 in Malaysia [29]; 1995–1997 in India [62]; 1996–2004 in India [63]; 2006–2008 in Vietnam [141]; 2009–2010 in India [100]; 2010–2013 in India [101]; 2011 in India [143]; 2016 in India [159]	1978–1980 in Indonesia [57]; 1982 in Thailand [135]; 1991–1994 in India [53]; 2010 in India [160]	[105]

* Vector competence was proven by experimental infection of mosquitoes and successful transmission (to an animal or JEV detected in saliva/salivary glands).

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
