# Peer review of "Mosquito Vector Competence for Japanese Encephalitis Virus"

_viruses, 2021, doi:10.3390/v13061154_

Round 1

Reviewer 1 Report

In the review from Auerwald et. al., the authors provide a very comprehensive overview with regard to the mosquito vector competence for Japanese Encephalitis virus. The authors listed and described results obtained from field-caught mosquitoes (virus detection and/or virus isolation) as well as from experimentally infected mosquitoes. Overall, this is a nice and detailed review. However, the classification into ‘the authors’ categories is not always really clear since in parts of the manuscript they divide into 2 categories: confirmed and potential. And in other parts they mention three categories (e.g. lines 131 -134) - e.g. in line 134 it is not clear whether category (iii) will belongs also to ‘potential vectors’.

Based on the tables it seems ‘confirmed vectors’ is the case when positive results for field-caught AND experimentally infected mosquitoes are available. ‘Potential vectors’ is the case when data are available from EITHER field-caught OR experimentally infected mosquitoes. If this is the case, please rephrase or restructure accordingly throughout the manuscript (also make it more clear in the Abstract – similar to eg. ilines 148/149 where the potential vector category is described nicely). Restructuring also includes the headings within the results part. Here I would suggest to divide into:

3.1. Confirmed vectors

3.2. Potential vectors

3.2.2 JEV detection and/or isolation from field-caught mosquitos (or similar)

3.2.3 Vector competence studies under experimental conditions (or similar)

As listed in my suggestion, for the potential vector section I would fist describe the field-caught results and then the experimentally results (thus the other way around as you did). First, this fits better to the order in Table 2 and second – having the experimental conditions in the second part gives a nicer transition to table 3 which lists the results of experimental vector competence studies.  

Additional comments:

Line 14/ Abstract:

Japanese encephalitis (JE) is the zoonosis (not JEV) – JEV is the agent of the zoonosis. E.g. check also WHO website. Please change accordingly.

Line 17: at this point the definition for confirmed vector competence is not clear because 'documented vector competence' is not defined at that point. It would include mosquitoes only with positive experimental infections?

Lines 17-18: sentence not very clear / word missing? And JEV was found?

Or rephrase e.g.: vector competence was confirmed for 14 species of wild mosquitoes. 

Line 20: five genera (instead of five genus)

Line 36: ‘JE’ is double

Line 41 – delete comma for pig-to-pig

Line 43: ? - for the maintenance (instead of for the maintained) ?

Lines 49/50 and 65/66: little confusing: is now rather Cx quinquefasciatus or vishnui a major/primary vector?

Line 57: south? at least west and east was also in small letter. Or West and East also in capital

Line 85: dot missing at end of sentence

Lines 86 ff: sentence very long and kind of confusing. Either separate in several sentences or alternatively make kind of enumeration: categorize in (i) potential vector when… (ii) confirmed vectors

Line 101: according to

Lines 102/103? Just wondering: What other studies could there be? What was excluded for example?

Section 2: Materials and Metods

Inlcude subheadings in Material and Methods. E.g.

  1. Literature search,
  2. Classification and mosquito taxonomy
  3. Vector competence definition/or indices? (or similar)

Lines 128/129: rephrase – because it sounds like the excluded once demonstrate JEV detection and/or isolation …

Line 132: proven by? Experimental infection?

Lines 133 and 134 – is there a difference between ‘found’ in field-caught mosquitoes and ‘documented’ JEV isolation? Do the mosquitos of point (iii) belong to confirmed or potential? Here it gets confusing again with regard to definition: confirmed versus potential.

Figure 2: is it only a specific (or limited number of) mosquito species for Italy? If so, it would be nice to mention it/them in the figure legend.

Table 1: Proven vector competence = after experimentally feeding? Please specify in food note or directly mention in heading.

Line 151: seven different mosquito genera

Line 157 – was isolated – can this be specified? In cell culture? Using which cells?

Line 165/166: rephrase: …in A. albopictus infected either by intrathoracic injection or by feeding an infectious …

Line 193: several times (plural)

Line 214: by feeding instead of due to feeding

Line 226: JEV was detected in Cx. collected 1987-88 in Sri Lanka

Line 251: by detection of or by detecting

Line 256: delete ‘was’ after transmission > transmission to mice

Line 273: with a sufficiently high (not with sufficiently a high)

Line 274: lizards (plural)

Line 298: lizards (plural)

Line 310: please specify/name virus family of Bagaza virus

Line 333: virus isolation was also successful (instead of attempts were successful)

Line 378: five genera

Lines 380-382: the list only contains 11 species but in line 378 it says 13 species. And in table 2 only 12 species were listed for ‘proven vector competence’ (which also should be the once tested in laboratory, right).

Line 390: This finding is lower that????

Line 395: sixteen days (plural)

Line 400: Ae. japonicus in italics

Line 411: observed with a mosquito? Means in one mosquito?

Line 449: delete 1x not

Table 3 is very lengthy and makes the manuscript very long. Move Table 3 to supplements.

Line 481: what is the ‘ highest minimum infection rate’?

Line 485: Which is in accordance with – (instead of that are in accordance)

Line 487: and is certainly studied the most

Line 508: ? often underlined participating in emergence?? > What do you want to say?  Rephrase, make two sentences... grammar?

Line 534: in this review we identified Ae…… as confirmed vectors. (complete the sentence with this addition).

Line 582: besides this initial study

Line 609ff: make new sentence - and as the impact …

Different feeding methods were discussed in the discussion: can one say whether feeding on animals or feeding with ‘artificial’ blood meal worked better? If so, please add information to discussion.

Author Response

We thank the reviewer for their thoughtful comments and efforts towards improving our manuscript. To avoid confusion regarding the category of potential JEV vector species, we re-structured the manuscript accordingly. As we considered potential vector species as mosquitoes with proven vector competence in the laboratory but lacking evidence of JEV presence in field caught mosquitoes, we deleted the potential vector species from table 2 and only left species with solely JEV detection/isolation from the field in this table. Regarding the exclusion of certain studies: the majority of the excluded publications only described the abundance of JEV vector species and the authors assumed their status as JEV vectors but did not prove the presence of the virus itself.

We refrained from giving details on:

  • detailed JEV vector distribution in Italy as it exceeds the scope of this review. The countries highlighted in figure 2 indicate JEV detection and isolations from field-caught mosquitoes of one or several species considered as confirmed by our criteria.
  • details on the isolation of JEV as this varies greatly, especially as in the 1970s-80s isolation by intracranial injection into suckling mice was the preferred method, whereas nowadays isolation with established cell lines is nearly exclusively used.

We agree with the reviewer that table 3 enlarges the manuscript dramatically. We await a recommendation from the editors whether to move it to the supplements or not.

Reviewer 2 Report

The authors performed an extensive search of the literature for JEV vectors to generate a list of probable vectors of JE.  The data from the studies were sorted by whether the mosquito was found infected in the field, or was laboratory infected, including if virus is found in the body (infection) or in saliva/ or salivary glands (transmission). The authors identified 36 vectors implicated as vectors of JEV based on field studies and laboratory vector competence studies. Results from the study point out the issues characterizing a specific mosquito species as a vector because of the number of different methods used to assess vector ability and supports adopting standardized methods, especially those used for transmission verification. 

Some minor English editing is needed; Lines 36, 193, 256.

Author Response

We thank the reviewer for their positive comment. We corrected the mentioned errors.

Reviewer 3 Report

see attachment

Author Response

We thank the reviewer for their positive comments and careful review, which helped improve the manuscript. We implemented the suggestions and corrected the errors. We decided to keep the repetition of the criteria for confirmed and potential vector species at the beginning of the results section, as we believe that reviews are often not read completely and the repetition would make it easier for readers who directly jump to the results section.